# From the dual cyclone harvest performance of single conidium powder to the effect of *Metarhizium anisopliae* on the management of *Thaumastocoris peregrinus* (Hemiptera: Thaumastocoridae)

Simone Graziele Moio Velozo[1,2], Murilo Rodrigues Velozo[3], Maurício Magalhães Domingues[1]*, Luciane Katarine Becchi[1], Vanessa Rafaela de Carvalho[1], José Raimundo de Souza Passos[4], José Cola Zanuncio[5], José Eduardo Serrão[2], Dietrich Stephan[6], Carlos Frederico Wilcken[1]

1 Faculdade de Ciências Agronômicas, Laboratório de Controle Biológico de Pragas Florestais, Universidade Estadual Paulista (UNESP), Botucatu, São Paulo, Brasil, 2 Departamento de Biologia Geral, Universidade Federal de Viçosa, Viçosa, Minas Gerais, Brasil, 3 Syntech Research, Piracicaba, São Paulo, Brasil, 4 Departamento de Bioestatística, Instituto de Biociências, Universidade Estadual Paulista (UNESP), Botucatu, São Paulo, Brasil, 5 Departamento de Entomologia/BIOAGRO, Universidade Federal de Viçosa, Viçosa, Minas Gerais, Brasil, 6 Julius Kühn-Institut, Institute for Biological Control, Federal Research Centre for Cultivated Plants, Darmstadt, Germany

* mauricio.m.domingues@unesp.br

## Abstract

Insect pests introduced in eucalyptus plantations in Brazil are mostly of Australian origin, but native microorganisms have potential for their management. High quality biopesticide production based on entomopathogenic fungi depends on adequate technologies. The objective of this study was to evaluate Mycoharvester® equipment to harvest and separating particles to obtain pure *Metarhizium anisopliae* conidia to manage *Thaumastocoris peregrinus* Carpintero & Dellapé, 2006 (Hemiptera: Thaumastocoridae). The Mycoharvester® version 5b harvested and separated *M. anisopliae* spores. The pure conidia were suspended in Tween 80® (0.1%) and calibrated to the concentrations of $1 \times 10^6$, $10^7$, $10^8$ and $10^9$ conidia/ml to evaluate the pathogenicity, lethal concentration 50 and 90 ($LC_{50}$, $LC_{90}$) and lethal time 50 and 90 ($LT_{50}$, $LT_{90}$) of this fungus to *T. peregrinus*. This equipment harvested 85% of the conidia from rice, with production of $4.8 \pm 0.38 \times 10^9$ conidia/g dry mass of substrate + fungus. The water content of 6.36% of the single spore powder (pure conidia) separated by the Mycoharvester® was lower than that of the agglomerated product. The product harvested at the concentrations of $10^8$ and $10^9$ conidia/ml caused high mortality to *T. peregrinus* third instar nymphs and adults. The separation of conidia produced by solid-state fermentation with the Mycoharvester® is an important step toward optimizing the fungal production system of pure conidia, and to formulate biopesticides for insect pest management.

**Data Availability Statement:** All data generated or analyzed during this study are included in this paper.

**Funding:** Funding was provided by the following Brazilian institutions: "Conselho Nacional de Desenvolvimento Científico e Tecnológico (CNPq)", Coordenação de Aperfeiçoamento de Pessoal de Nível Superior – Brazil (CAPES) granted the the scholarship in Germany (Financing Code 88881.134760/2016-01) and in Brazil (Financing Code 001) to S.G.M.V. and Programa Cooperativo sobre Proteção Florestal/PROTEF do Instituto de Pesquisas e Estudos Florestais/IPEF. The funders had no role in study design, data collection and analysis, decision to publish, or preparation of the manuscript.

**Competing interests:** The authors have declared that no competing interests exist.

## Introduction

Planted forests contribute to the Brazilian economy with 9.93 million hectares, of which 75.8% are *Eucalyptus* spp. [1]. Exotic insect pests such as *Thaumastocoris peregrinus* Carpintero & Dellapé, 2006 (Hemiptera: Thaumastocoridae), *Glycaspis brimblecombei* Moore, 1964 (Hemiptera: Aphalaridae) and *Gonipterus platensis* Marelli, 1926 (Coleoptera: Curculionidae) threaten these numbers [2–4]. The dispersion and damages by *T. peregrinus* on different *Eucalyptus* species plantations are high [5,6].

Monitoring is essential to determine *T. peregrinus* egg, nymph and adult numbers in the plant canopy and to manage this pest [7]. The parasitoid *Cleruchoides noackae* Lin & Huber, 2007 (Hymenoptera: Mymaridae) [7], insect predators [8,9] and entomopathogenic fungi are important to manage *T. peregrinus* and other forest insects [10–12]. However, the production of these microorganisms in developing countries requires intense manual labor, which can reduce the quantity and quality of the final product [13,14]. This increases the difficulties of obtaining large quantities of biopesticides of acceptable quality to meet the requirements of forest and agricultural areas.

A total of 171 mycoinsecticides were produced with at least 12 of the more than 800 species of entomopathogenic fungi, emphasizing the genera *Beauveria*, *Cordyceps* and *Metarhizium*, with formulations in the form of substrates colonized by fungi, wettable powders and dispersions in oil being the most common [15]. The dual cyclone technology harvest conidia with advantages of eliminating large particles, a high-quality spore separation, operator safety, economical convenience processing, ease of storage and high conidia productivity, such as for *Beauveria bassiana* and *Metarhizium anisopliae* [16]. This technology needs further development for using in the formulation and application of mycoinsecticides, contributing directly to the integrated management of agriculture and forest pests. The objective of this study is to evaluate harvesting technology with particle separation to obtain pure conidia of *Metarhizium anisopliae* (IBCB425), and to evaluate the pathogenicity of the product obtained against *T. peregrinus*.

## Materials and methods

### Obtaining the fungus by solid fermentation

The fungus *Metarhizium anisopliae sensu lato* (Hypocreales: Clavicipitaceae) (IBCB425) was isolated from Atlantic Forest soil collected in Iporanga, São Paulo state, Brazil and deposited in the Oldemar Cardim Abreu Entomopathogenic Microorganism Collection at the "Centro Avançado de Pesquisa em Proteção de Plantas e Saúde Animal" (Advanced Center for Research in Protection of Plant and Animal Health) in Campinas, São Paulo, Brazil. Large quantities of the IBCB425 (strain) conidia were obtained with a solid-state fermentation process at the Julius Kühn-Institute, Darmstadt, Germany. The ProPhyta L03 Fermenter System® [Prophyta, Germany] laboratory bioreactor was filled with a substrate of parboiled rice[a] + barley[b], 4:1 (160g[a] + 40g[b]) to mass-produce this fungus at 25˚C with an aeration rate of 150 L/h [17]. The *M. anisopliae* conidia were harvested after 14 days of fermentation and one day for extraction and quantification.

### Obtaining *Thaumastocoris peregrinus*

Branches infested with *T. peregrinus* adults were collected from *Eucalyptus grandis* × *E. urophylla* hybrid plants and taken to the Laboratory of Biological Control of Forest Pests (LCBPF), of the Department of Plant Protection, "Faculdade de Ciências Agronômicas, Universidade Estadual Paulista Júlio de Mesquita Filho—FCA, UNESP" (College of Agronomic

Sciences, São Paulo State University) in Botucatu, São Paulo, Brazil to mass rear this insect. Bouquets were formed with branches of the hybrid *Eucalyptus urophylla* var. *platyphylla*, supported on a 250 ml Erlenmeyer flask with water in a rectangular plastic tray (40 cm long × 35 cm wide × 8 cm high), and were replaced every three to four days, placing the oldest and driest branches next to the new ones to facilitate *T. peregrinus* migration [18]. Adults and third instar nymphs were used in the pathogenicity assay.

### Product harvest and Mycoharvester® system

The Mycoharvester® system [Mycoharvester Company, UK] separates pure conidia and agglomerate from fungus-colonized substrates using double cyclone technology [19]. A total of 150 g of the substrate + fungus per treatment and replication was placed inside the Mycoharvester® to harvest the conidia. The harvest quality was based on the conidia weight (grams), the number of conidia per gram and the percentage of humidity of the products obtained. In addition, the product obtained (substrate + fungus) was evaluated, both after the fermentation period and following harvesting, to determine the conidia number per gram of dry mass. Twenty-five grams of the substrate were washed with 225 ml Tween 80® [Sigma, Germany] (0.1%). The suspension was shaken for 10 minutes in a KS 15A Orbital Shaker compact agitator [Edmund Bühler GmbH, Germany] at 420 rpm to separate the conidia. The conidia on the substrate were counted before and after passing through the harvesting system. The conidia harvest period was one day. The experiment was repeated independently three times under the same conditions. Fungus production and harvesting experiments were carried out at the Julius Kühn-Institute, Darmstadt, Germany.

### Pathogenicity to *Thaumastocoris peregrinus*

The pure conidia with 98.0 ± 1.013% germination rate [20] were suspended in Tween 80® [Sigma, Germany] (0.1%) and their quantity calibrated to the concentrations of $1 \times 10^6$, $10^7$, $10^8$ and $10^9$ of conidia/ml to evaluate the pathogenicity, lethal concentration 50 and 90 ($LC_{50}$, $LC_{90}$) and lethal time 50 and 90 ($LT_{50}$, $LT_{90}$) of the fungus. Then, 2 ml of conidial suspension per concentration were sprayed using a Potter spray tower [Burkard, UK] (from lowest to highest concentration) on ten *T. peregrinus* adults and ten *T. peregrinus* third instar nymphs (separately), from the laboratory rearing stock for each of the three replications. The control received only Tween 80® (0.1%). The insects were kept in Petri dishes with their lids covered with voile fabric, allowing gas exchange, and with a *Eucalyptus urophylla* var. *platyphylla* leaf from six-year-old trees to feed the insects. These leaves were placed on a thin layer of hydroretentor gel (1 g/400 ml) [Hydroplan-EB, Empresa de Base & Distribuidora Ltda, Brazil] to maintain their turgescence and prevent the insects to escape. After applying the fungus on the insects, the Petri dishes were kept in bio-oxygen demand (BOD) incubators at a temperature of 25˚C ± 1 and photoperiod of 12h. Each replication (Petri dish) was observed daily for seven days, and dead insects were counted, removed and transferred to humidity chambers to confirm the fungus pathogenicity. The experiment had three replications with 10 nymphs and 10 adults (separately) each, and the tests were repeated twice, totaling 60 individuals per insect stage and treatment. The pathogenicity was evaluated at the LCBPF (FCA, UNESP).

### DNA extraction and PCR

DNA extraction and polymerase chain reactions (PCR) were conducted at the Laboratory of Molecular Biology and Nematology (FCA, UNESP). Mycelium and conidia of the IBCB425 isolate with 10 days growth on potato agar dextrose (PDA) culture medium [Kasvi, Italy] at 25˚C were used for molecular identification. The microtube containing the sample received

80 μl of 10% Chelex100Ⓡ resin solution [Bio-Rad Laboratories, USA] and 8 μl of proteinase K (20 mg/ml). Then, the microtube containing the sample was placed in a thermal block at 95˚C for 20 minutes [21]. The supernatant was collected for PCR. The PCR reaction for the amplification of the ITS1-5.8S-ITS2 region of rDNA was performed in a total volume of 50 μL, using Taq DNA polymerase 1X buffer, 1.5 mM MgCl2, 0.4 μM each of ITS1 (5'-TCCGTAGGT−GAACCTGC GG-3') and ITS4 (5'-TCCTCCGCTTATTGATATGC-3') primers, 0.2 mM dNTPs, 0.2 U of Taq DNA polymerase and 25ng of DNA. Amplification was conducted in a thermocycler [Infinigen, TC-96CG, USA] programmed for initial denaturation at 95˚C for 5 min, followed by 30 cycles of 95˚C denaturation for 30 sec; annealing at 62˚C for 1 min; extension at 72˚C for 2 min; and final extension at 72˚C for 5 min.

The DNA extraction and PCR reaction (50 μL) products were visualized by 1% agarose gel electrophoresis under UV light and photodocumented. The amplification products were sent to IBTEC Central Laboratory of the UNESP/Botucatu, São Paulo state, Brasil and automated Sanger DNA sequencing was performed (Model: ABI 3500—Applied Biosystems).

The sequence obtained was used to find the most similar ones deposited in the GenBank, using the BLASTn tool. These sequences were aligned and compared in BLAST and deposited in the GenBank database (http://www.ncbi.nlm.nih.gov). Molecular analysis confirmed the fungus species at the beginning and at the end of the research.

## Statistical analysis

Generalized linear models with gamma distribution and logarithmic link function were adjusted for the number of spores per gram before and after harvest, considering a structure of measures repeated by the Generalized Estimation Equation method [22–24]. The first considered the conidia mass in grams as a product type factor and the blocks as covariables; the second, water content, considered the product type as factor and the block as covariable; and the third considered the death rate per insect stage as a factor, and time and replication as continuous covariables with block as covariable [22]. The Lethal Concentrations ($LC_{50}$, $LC_{90}$) and Lethal Times ($LT_{50}$, $LT_{90}$), followed by the confidence interval, were obtained by fitting a logistic regression model with binomial distribution and probit link function [22]. The procedure used was GENMOD (from the SAS Statistical Software, SAS University Edition), and the Tukey-Kramer test to compare treatments [25]. The graphs were plotted with SigmaPlot 12.0 software.

## Ethics statement

No specific permits are required to rear *T. peregrinus* in Brazil. The services to produce and harvest the fungus were executed in Germany based on a legal document pursuant to § 6 Article 24 of Decree No. 8.772, dated May 11[th], 2016, regulated by the National Law No. 13,123 of May 20[th], 2015, providing access to genetic heritage, protection and access to associated traditional knowledge and the sharing of benefits for the conservation and sustainable use of biodiversity in Brazil. This was signed between the UNESP/FCA and the Julius Kühn Institute. The registration of the project was carried out in SISGEN. The isolate IBCB425 was returned to Brazil at the end of this research. The laboratory studies did not involve endangered or protected species in Brazil or Germany.

# Results

## Molecular analysis

The sequence obtained in the molecular analysis showed coverage and high identity, with 100% for the species *Metarhizium anisopliae* (GenBank accession number MT278261.1).

## Product harvest and the Mycoharvester® system

The *M. anisopliae* yield was $4.8 \pm 0.38 \times 10^9$ conidia/g of dry mass of substrate + fungus with water content of $57.7 \pm 1.06\%$. A total of 150 g of substrate colonized by $4.8 \pm 0.38 \times 10^9$ conidia/g of the fungus provided $3.12 \pm 0.21$ g of pure conidia with $5.31 \pm 0.28 \times 10^{10}$ conidia/g, and $1.14 \pm 0.09$ g of the agglomerate with $1.16 \pm 0.24 \times 10^{10}$ conidia/g (Fig 1A). The word "agglomerate" means a product with conidia, mycelia and residues from the substrate. The substrate + fungus yield in the Mycoharvester® version 5b (Fig 2A) before and after (Figs 1B and 2B and 2C) using this system was $4.8 \pm 0.38 \times 10^9$ conidia/g and $6.9 \pm 0.71 \times 10^8$ conidia/g dry mass, respectively, confirming that this method collects approximately 85% of the conidia with approximately 73% is pure (Figs 1A and 2D). The water content of the products differed, with $6.36 \pm 0.13\%$ for pure conidia and $34.84 \pm 0.65\%$ for the agglomerate (Fig 1C).

## Pathogenicity to *Thaumastocoris peregrinus*

The pure conidia of *M. anisopliae* (IBCB425), applied at different concentrations, was pathogenic to third instar nymphs and adults of *T. peregrinus*, being more virulent at the concentrations of $10^8$ and $10^9$ conidia/ml ($P < 0.05$). Averages of all treatments differed from that of the control ($P < 0.05$), except the $10^6$ conidia/ml concentration applied to nymphs and to nymphs and adults together (Fig 3 and Table 1).

Visible conidiogenesis and fungus sporulation was observed on the bodies of *T. peregrinus* third instar nymphs and adults killed by *M. anisopliae* (IBCB425) (Fig 4). This fungus, at the two highest concentrations, applied on day zero, penetrated the insects on the first and second day, killed them on the second to fifth day, sporulated on the fourth to seventh day, and thereafter covered them with the fungal mass.

The 50 and 90 lethal times of the IBCB425 isolate were lower with the higher concentrations ($10^8$ and $10^9$ conidia/ml) needing 4.85 (4.53–5.28) and 3.22 (3.04–3.43) days and 7.37 (6.69–8.37) and 4.61 (4.27–5.09) days, respectively, to kill 50% and 90% of the *T. peregrinus* population (Table 2). The lethal concentration to kill 50% of the population was, respectively, $6.4 \times 10^7$ ($2.7 \times 10^6$–$1.6 \times 10^7$) and $1.5 \times 10^9$ ($5.5 \times 10^8$–$5 \times 10^9$) conidia/ml.

# Discussion

The molecular analysis confirmed that the IBCB425 has been reported and recorded in Brazil as *M. anisopliae* [26–28].

Production in a solid bioreactor generated significant quantities of pure conidia of *M. anisopliae* (IBCB425). The production of this fungus in solid-state fermentation with rice as a substrate is around $10^9$ conidia/g, although the optimization of factors such as rice moisture between 22 and 30% can increase these values, demonstrating the efficiency of this method [29,30]. The quality of *M. anisopliae* production in a solid-state bioreactor with $3.4 \times 10^9$ conidia/g obtained with rice bran was the highest in a test using different substrates in aerated packed beds [31].

The high yield of conidia collection and separation in the Mycoharvester® is similar to that reported using this technique for *B. bassiana* and *M. anisopliae*, with 12.6 mg/g and $1.8 \times 10^{10}$ conidia/g, with yield superior to that from using sieving and washing methods with

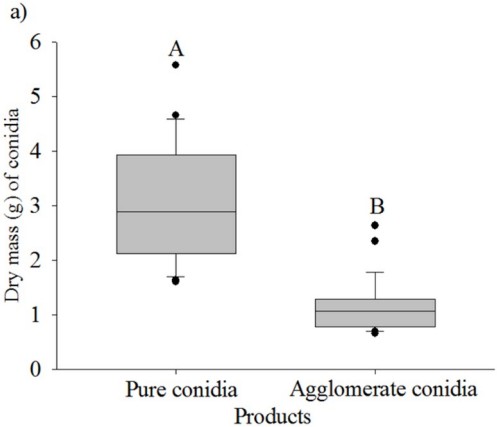

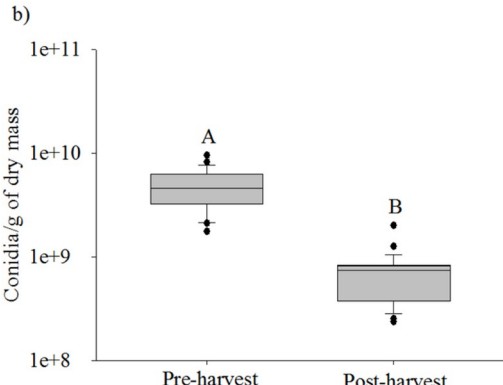

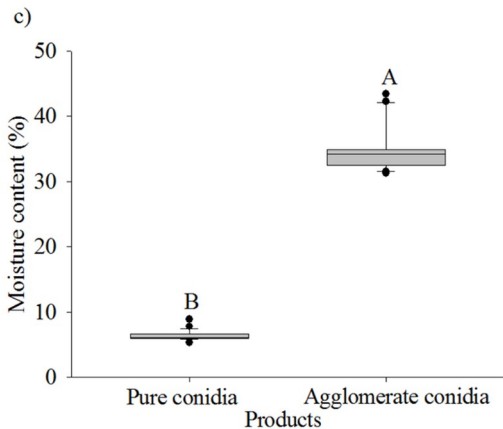

**Fig 1. Evaluation of conidia harvest with Mycoharvester® System.** a: Dry mass (g) of the pure and agglomerate conidia in 150g of colonized substrate (P< 0.0001). b: Water content of the pure and agglomerate conidia of *Metarhizium anisopliae* after harvesting the material (P < 0.0001). c: Performance of Mycoharvester® version 5b in the conidia harvest of *Metarhizium anisopliae*: Comparison of the substrate + fungus before and after the harvest procedure (P< 0.0001). Means followed by the same capital letter are similar by the Tukey-Kramer test at 5% significance level.

Tween 80 (0.02%) [16]. Aerial conidia is the principal and most effective infectious stage of the entomopathogenic Hypocreales [32] and the ease of production of pure conidium in the

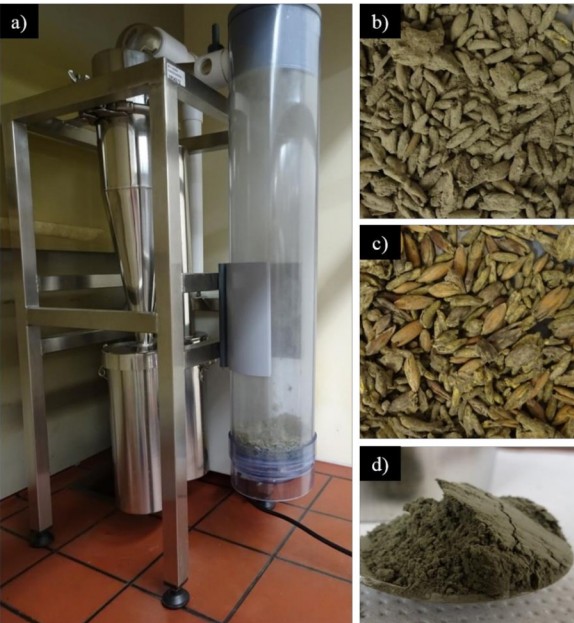

**Fig 2. Mycoharvester® System.** Mycoharvester® equipment, version 5b (a); substrate colonized by *Metarhizium anisopliae* before (b) and after harvest (c); pure conidia of *Metarhizium anisopliae* (d).

technique studied indicates its potential for large-scale production and application. Mycoharvester® improved the harvesting of pure conidia without using water, which may impair the stability and purity of the product, to develop new oily formulations based on *Metarhizium* spp. to control locusts in Africa [33].

The water content of the products, 6% for pure conidia and 34% for the agglomerate, shows the adequacy of the Mycoharvester® for conidia separation in developing biopesticides based on entomopathogenic fungi. The water content of conidia from solid substrates must be lower than 9% for an adequate shelf life, regardless of whether the product is formulated [13,32,34]. The water content of the conidia affects its germination during storage periods, and relative humidity of 2.5% to 6.2% was ideal for the high germination of *Metarhizium flavoviride* conidia, even after exposed to conditions of 50˚C [35]. However, dry infective propagules, immersing in water for rehydration can severely damage plasma membranes, which is usually lethal for single-celled organisms [36]. This occurred for dried conidia of *M. anisopliae* suspended in water at 0.5˚C and 33˚C, with germination rates of 0.9% and 94%, respectively, when rehydrated, showing the importance of care during this process [36].

The high *T. peregrinus* mortality by the *Metarhizium anisopliae* (IBCB425) indicates its potential to managing this pest with its conidia applied in suspension. This is similar to that reported for different isolates of this fungus, including the IBCB425 and IBCB348, with mortality of up to 100% of other Hemiptera, such as the sugarcane spittlebug *Mahanarva fimbriolata* Stål (Hemiptera: Cercopidae) [27], and adults of *Diaphorina citri* Kuwayama, 1908 (Hemiptera: Liviidae) in the laboratory [37]. Native entomopathogenic fungi can improve control of exotic forest pests in sustainable management programs, as found for successful cases of efficiency by fungi of natural occurrence such as *B. bassiana*, *Cordyceps* sp. and *Zoophtora radicans* [11,38,39].

The isolate IBCB425 was more virulent to *T. peregrinus* at the highest concentrations, requiring $6.4 \times 10^7$ ($2.7 \times 10^7$–$1.6 \times 10^8$) and $1.5 \times 10^9$ ($5.5 \times 10^8$–$5 \times 10^9$) conidia/ml to control

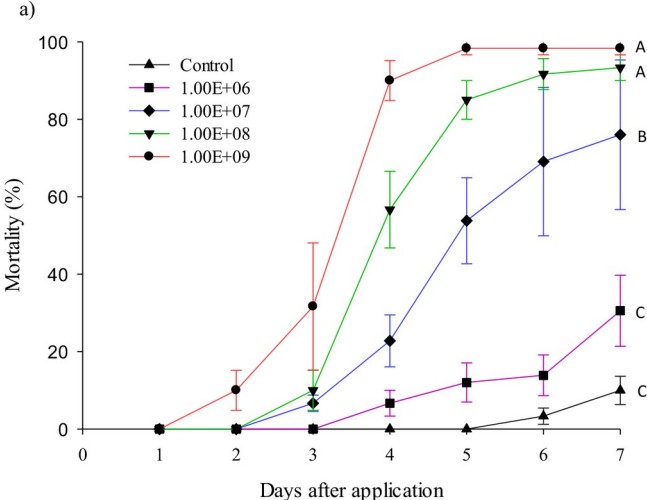

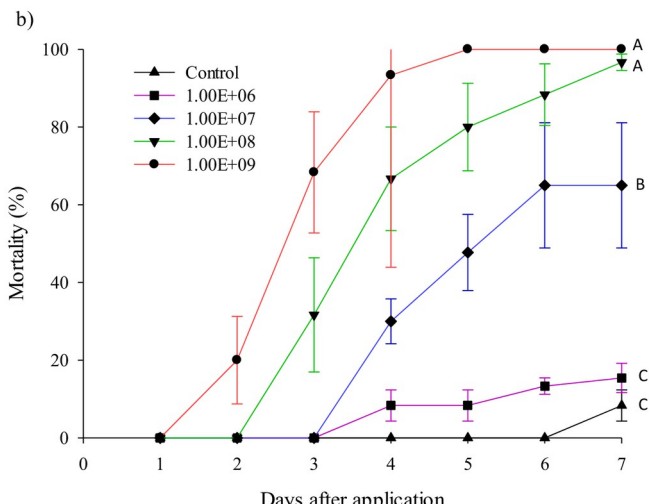

**Fig 3.** Cumulative pathogenicity of *Metarhizium anisopliae* at different concentrations of pure conidia to adults (a) and third instar nymphs (b) of *Thaumastocoris peregrinus* (Hemiptera: Thaumastocoridae) seven days after its application. Means followed by the same capital letters are similar by the Tukey-Kramer test at the 5% significance level.

**Table 1. Mortality of third instar nymphs and adults of *Thaumastocoris peregrinus* (Hemiptera: Thaumastocoridae) by *Metarhizium anisopliae* (IBCB425) (%) in pure conidia concentrations (conidia/mL).**

| IBCB425 | Nymphs | Adults | Total |
|---|---|---|---|
| Control | 8.33 ± 4.01 C | 10.00 ± 3.65 C | 9.17 ± 2.60 C |
| $10^6$ | 15.00 ± 3.42 C | 28.33 ± 8.33 B | 21.67 ± 4.74 C |
| $10^7$ | 50.00 ± 10.00 B | 55.00 ± 8.85 B | 52.50 ± 6.41 B |
| $10^8$ | 96.67 ± 2.11 A | 93.33 ± 3.33 A | 95.00 ± 1.95 A |
| $10^9$ | 100.00 ± 0.00 A | 98.33 ± 1.67 A | 99.17 ± 0.83 A |

Means followed by the same capital letter per column do not differ by the Tukey-Kramer test (Westfall, et al., 1999) at 5% significance.

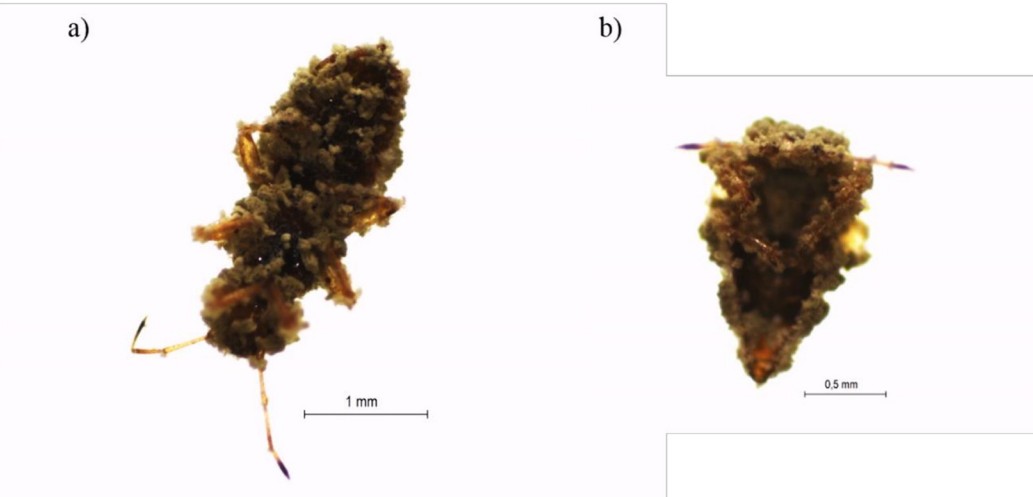

**Fig 4.** Adult (a) and third instar nymph (b) of *Thaumastocoris peregrinus* (Hemiptera: Thaumastocoridae) colonized by *Metarhizium anisopliae.*

50 and 90%, respectively of this insect at both stages (nymphs and adults), which is $10^{11}$ to $10^{13}$ conidia/ha. These values make it feasible to use entomopathogenic fungi in field conditions against *T. peregrinus*, since products are generally commercialized and recommended at the field concentration of 1 x $10^{12}$ to 1 x $10^{13}$ conidia/ha [40–42]. The concentration of 5 x $10^{12}$ conidia/ha controlled *D. citri* up to 95.6% (*B. bassiana*) and 94% (*Cordyceps fumosorosea*) under field conditions [42]. Commercial products based on *B. bassiana* and *M. anisopliae* at a concentration of 1 x $10^8$ conidia/ml caused 100 and 88% mortality of *T. peregrinus* adults, respectively, after 11 days under laboratory conditions [43], similar to that of the commercial isolate tested in our bioassays. The $LC_{50}$ of *B. bassiana* blastospores, produced in a submerged medium, was higher than that of the IBCB425, approximately 3 x $10^7$ conidia/ml for three different isolates, while it did not differ from the $LC_{50}$ of aerial conidia [44]. Different isolates of *M. anisopliae*, at concentrations of $10^5$, $10^6$, $10^7$ and $10^8$, were pathogenic at the highest concentrations to *Rhipicephalus* (*Boophilus*) *microplus* Canestrini, 1888 (Acari: Ixodida) [45]. The concentration of 1.2 x $10^7$ conidia/ml of *M. anisopliae* (IBCB348) caused high *M. fimbriolata* mortality [46]. Formulations based on *B. bassiana*, *M. anisopliae* and *Lecanicillium lecanii* were pathogenic to *G. brimblecombei*, with increased mortality as the conidia concentration increased [10].

The use of native entomopathogenic fungi to manage introduced forest pests increases the viability of sustainable programs. Naturally occurring entomopathogenic fungi such as *B. bassiana*, *Cordyceps* spp., *Beauveria pseudobassiana* and *Zoophtora radicans* are common in

**Table 2. Lethal times (50 and 90) of *Metarhizium anisopliae* (IBCB425) (days) with different concentrations (conidia/mL) in the mortality of *Thaumastocoris peregrinus* third instar nymphs and adults (Hemiptera: Thaumastocoridae).**

| IBCB425 | Lethal time 50 | Lethal time 90 |
|---|---|---|
| Control | 8.27 (7.63–11.30) | 9.40 (8.33–14.73) |
| $10^6$ | 11.42 (9.38–17.14) | 15.71 (12.31–25.48) |
| $10^7$ | 10.65 (8.75–15.06) | 16.97 (13.28–25.75) |
| $10^8$ | 4.85 (4.53–5.28) | 7.37 (6.69–8.37) |
| $10^9$ | 3.22 (3.04–3.43) | 4.61 (4.27–5.09) |

the field, and these species show promise against *T. peregrinus* [38,39,47]. The mortality of this insect, from 80.1% to 100% with *B. bassiana*, 88% with *M. anisopliae*, and 87% with *Cordyceps* sp. at a concentration of $10^8$ conidia/ml, and 98% with *B. bassiana* at $10^7$ conidia/ml demonstrates the potential of these microorganisms to manage this pest [12,43,47]. Successful cases in the agricultural sector with entomopathogenic fungi are reported, especially with *M. anisopliae*, for decades, against Hemiptera pests in sugarcane cultures in Brazil [14]. The high pathogenicity of the final product obtained with the dual cyclone harvest technique using Mycoharvester, with results similar to formulated products efficient against insect pests, makes possible to collect and produce pure conidia without interfering with entomopathogenic fungus pathogenicity. This is important to improve and develop mycoinsecticides and highlights the relevance of this still poorly investigated technology.

The Mycoharvester® equipment is highly efficient to collect, separate and to obtain high quality of pure *M. anisopliae* conidia. The pure conidia obtained with this technology shows potential to manage *T. peregrinus* adults and nymphs.

## Acknowledgments

Dr. Phillip John Villani (University of Melbourne, Australia) revised and corrected the English language used in this manuscript.

## Author Contributions

**Conceptualization:** Simone Graziele Moio Velozo, Maurício Magalhães Domingues, Vanessa Rafaela de Carvalho, Dietrich Stephan, Carlos Frederico Wilcken.

**Data curation:** Simone Graziele Moio Velozo, Maurício Magalhães Domingues, José Raimundo de Souza Passos.

**Formal analysis:** Simone Graziele Moio Velozo, Maurício Magalhães Domingues, Vanessa Rafaela de Carvalho, José Raimundo de Souza Passos, Carlos Frederico Wilcken.

**Investigation:** Simone Graziele Moio Velozo, Murilo Rodrigues Velozo, Maurício Magalhães Domingues, Vanessa Rafaela de Carvalho, José Raimundo de Souza Passos, Dietrich Stephan.

**Methodology:** Simone Graziele Moio Velozo, Murilo Rodrigues Velozo, Luciane Katarine Becchi, Vanessa Rafaela de Carvalho, Dietrich Stephan, Carlos Frederico Wilcken.

**Project administration:** Simone Graziele Moio Velozo, Luciane Katarine Becchi, Vanessa Rafaela de Carvalho, Dietrich Stephan, Carlos Frederico Wilcken.

**Supervision:** Simone Graziele Moio Velozo, Carlos Frederico Wilcken.

**Validation:** Simone Graziele Moio Velozo, Murilo Rodrigues Velozo, Maurício Magalhães Domingues, Luciane Katarine Becchi, José Raimundo de Souza Passos, José Cola Zanuncio, José Eduardo Serrão, Dietrich Stephan, Carlos Frederico Wilcken.

**Visualization:** Simone Graziele Moio Velozo, Murilo Rodrigues Velozo, Maurício Magalhães Domingues, Luciane Katarine Becchi, Vanessa Rafaela de Carvalho, José Raimundo de Souza Passos, José Cola Zanuncio, José Eduardo Serrão, Dietrich Stephan, Carlos Frederico Wilcken.

**Writing – original draft:** Simone Graziele Moio Velozo, Murilo Rodrigues Velozo, Maurício Magalhães Domingues, Vanessa Rafaela de Carvalho, José Raimundo de Souza Passos, José Cola Zanuncio, José Eduardo Serrão, Dietrich Stephan, Carlos Frederico Wilcken.

**Writing – review & editing:** Simone Graziele Moio Velozo, Murilo Rodrigues Velozo, Maurício Magalhães Domingues, Vanessa Rafaela de Carvalho, José Raimundo de Souza Passos, José Cola Zanuncio, José Eduardo Serrão, Carlos Frederico Wilcken.

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
