## [Decision Letter · Decision Letter 0]

26 Dec 2022

PONE-D-22-33367From the dual cyclone harvest performance of single conidium powder to the effect of Metarhizium anisopliae on the management of a forest pestPLOS ONE

Dear Dr. Domingues, 

Thank you for submitting your manuscript to PLOS ONE. After careful consideration, we feel that it has merit but does not fully meet PLOS ONE’s publication criteria as it currently stands. Therefore, we invite you to submit a revised version of the manuscript that addresses the points raised during the review process.

ACADEMIC EDITOR: The reviewers recommend minor revision of this manuscript. 

We look forward to receiving your revised manuscript.

Kind regards,

Shawky M Aboelhadid, PhD

Academic Editor

PLOS ONE

Journal Requirements:

"Funding was provided by the following Brazilian institutions: “Conselho Nacional de Desenvolvimento Científico e Tecnológico (CNPq)”, Coordenação de Aperfeiçoamento de Pessoal de Nível Superior – Brazil (CAPES) granted the the scholarship in Germany (Financing Code 88881.134760/2016-01) and in Brazil (Financing Code 001) to S.G.M.V. and Programa Cooperativo sobre Proteção Florestal/PROTEF do Instituto de Pesquisas e Estudos Florestais/IPEF. "

4. Please amend your authorship list in your manuscript file to include author José Eduardo Serrão.

Reviewers' comments:

Reviewer's Responses to Questions

**Comments to the Author**

1. Is the manuscript technically sound, and do the data support the conclusions?

Reviewer #1: Yes

2. Has the statistical analysis been performed appropriately and rigorously? 

Reviewer #1: Yes

3. Have the authors made all data underlying the findings in their manuscript fully available?

Reviewer #1: Yes

4. Is the manuscript presented in an intelligible fashion and written in standard English?

Reviewer #1: Yes

5. Review Comments to the Author

Reviewer #1: Dear Editor,

the manuscript presents two hypotheses: (a) harvesting technology (Mycoharvester) can be applied for fungal Brazilian isolate (IBCB425), (b) the pure conidia obtained by this technology is effective to manage T. peregrinus. Both technologies can help in the management of T. peregrinus.

The manuscript is well structured, however, in the generated pdf the results appear before the material and methods.

The few considerations are in the body of the text.

Congratulations to the authors.

6. PLOS authors have the option to publish the peer review history of their article (what does this mean?). If published, this will include your full peer review and any attached files.

Reviewer #1: No

While revising your submission, please upload your figure files to the Preflight Analysis and Conversion Engine (PACE) digital diagnostic tool, https://pacev2.apexcovantage.com/. PACE helps ensure that figures meet PLOS requirements. To use PACE, you must first register as a user. Registration is free. Then, login and navigate to the UPLOAD tab, where you will find detailed instructions on how to use the tool. If you encounter any issues or have any questions when using PACE, please email PLOS at figures@plos.org. Please note that Supporting Information files do not need this step.<quillbot-extension-portal></quillbot-extension-portal>

---

## [Author Response · Author response to Decision Letter 0]

11 Jan 2023

Special considerations for this manuscript:

This manuscript was submitted for its publication in PLOS ONE (PONE-D-22-33367) and were are returning it with the corrections requested by the Subject Editor as a completely revised version.

COMMENTS TO THE AUTHOR:

Subject Editor:

Journal Requirements:

A: The manuscript has been revised to PLOS ONE's style requirements.

A: Please consider the information indicated in the ‘Funding Information. The grant numbers have been revised. 

"Funding was provided by the following Brazilian institutions: “Conselho Nacional de Desenvolvimento Científico e Tecnológico (CNPq)”, Coordenação de Aperfeiçoamento de Pessoal de Nível Superior – Brazil (CAPES) granted the the scholarship in Germany (Financing Code 88881.134760/2016-01) and in Brazil (Financing Code 001) to S.G.M.V. and Programa Cooperativo sobre Proteção Florestal/PROTEF do Instituto de Pesquisas e Estudos Florestais/IPEF. "

A: We declare that: "The funders had no role in study design, data collection and analysis, decision to publish, or preparation of the manuscript." 

 4. Please amend your authorship list in your manuscript file to include the author José Eduardo Serrão.

A: The author has been included in the manuscript file.

A: We have added the full ethics statement in the ‘Methods’ section of our manuscript file. 

A: The reference list has been revised. No changes were necessary.

Review 1

The manuscript is well structured, however, in the generated pdf the results appear before the material and methods.

A: We would like to thank you for the critical reading, careful recommendations, and constructive criticism. We have accepted and addressed all suggestions. Our responses to the recommendations are addressed bellow in a point-by-point manner. 

L5- Specify to Thaumastocoris peregrinus (Hemiptera: Thaumastocoridae)

A: We adopted the suggestions provided by R1. 

L37- Which concentrations?

A: We adopted the suggestions provided by R1. 

L88- Following the order of presentation (according to the norms) here would be materials and methods

A: We adopted the suggestions provided by R1. 

L145- What stage of development?

A: We adopted the suggestions provided by R1. 

L286- Take a draft (timeline) that shows from the application in Potter spray to the progression of the fungus during the 7 days of evaluation. 

Or differents tratments in humidity chambers 

A: We adopted the suggestions provided by R1. We added the timeline in the results.

L288- which stages?

A: We adopted the suggestions provided by R1.

---

## [Decision Letter · Decision Letter 1]

24 Jan 2023

PONE-D-22-33367R1From the dual cyclone harvest performance of single conidium powder to the effect of Metarhizium anisopliae on the management to Thaumastocoris peregrinus (Hemiptera: Thaumastocoridae)PLOS ONE

Dear Dr. Domingues, 

Thank you for submitting your manuscript to PLOS ONE. After careful consideration, we feel that it has merit but does not fully meet PLOS ONE’s publication criteria as it currently stands. Therefore, we invite you to submit a revised version of the manuscript that addresses the points raised during the review process.

ACADEMIC EDITOR: According reviewer comments and as present in table 2; the mortality of control was higher than in concentrations 10 6 and 10 7. the reviewer comment is (It is curious that untreated insects show mortality faster than concentrations 10^6 and 10^7. Mortalities higher than 10% in control should have experiments redone").

We look forward to receiving your revised manuscript.

Kind regards,

Shawky M Aboelhadid, PhD

Academic Editor

PLOS ONE

Journal Requirements:

Reviewers' comments:

Reviewer's Responses to Questions

**Comments to the Author**

1. If the authors have adequately addressed your comments raised in a previous round of review and you feel that this manuscript is now acceptable for publication, you may indicate that here to bypass the “Comments to the Author” section, enter your conflict of interest statement in the “Confidential to Editor” section, and submit your "Accept" recommendation.

Reviewer #1: All comments have been addressed

2. Is the manuscript technically sound, and do the data support the conclusions?

Reviewer #1: Yes

3. Has the statistical analysis been performed appropriately and rigorously? 

Reviewer #1: Yes

4. Have the authors made all data underlying the findings in their manuscript fully available?

Reviewer #1: Yes

5. Is the manuscript presented in an intelligible fashion and written in standard English?

Reviewer #1: Yes

6. Review Comments to the Author

Reviewer #1: Dear editor,

The considerations requested in the first version were fulfilled, however, I added some comments, among them:

"Please review the mortality of control. It is curious that untreated insects show mortality faster than concentrations 10^6 and 10^7. Mortalities higher than 10% in control should have experiments redone".

7. PLOS authors have the option to publish the peer review history of their article (what does this mean?). If published, this will include your full peer review and any attached files.

Reviewer #1: No

While revising your submission, please upload your figure files to the Preflight Analysis and Conversion Engine (PACE) digital diagnostic tool, https://pacev2.apexcovantage.com/. PACE helps ensure that figures meet PLOS requirements. To use PACE, you must first register as a user. Registration is free. Then, login and navigate to the UPLOAD tab, where you will find detailed instructions on how to use the tool. If you encounter any issues or have any questions when using PACE, please email PLOS at figures@plos.org. Please note that Supporting Information files do not need this step.<quillbot-extension-portal></quillbot-extension-portal>

---

## [Author Response · Author response to Decision Letter 1]

3 Mar 2023

Special considerations for this manuscript:

This manuscript was submitted for its publication in PLOS ONE (PONE-D-22-33367R1) and we are returning it with the corrections requested by the Subject Editor as a completely revised version.

Reviewer 1

A: We would like to thank you for the critical reading, careful recommendations, and constructive criticism. We have accepted and addressed all suggestions. Our responses to the recommendations are addressed bellow in a point-by-point manner.

Reviewer #1: Dear editor,

The considerations requested in the first version were fulfilled, however, I added some comments, among them:

"Please review the mortality of control. It is curious that untreated insects show mortality faster than concentrations 10^6 and 10^7. Mortalities higher than 10% in control should have experiments redone".

A: For LT50, when separately comparing the confidence level for 10E6 and control, as well as for 10E7 and control, there is no significant statistic difference (p<0.05) between the means (confidence intervals intersect). The same applies to LT90. As for mortality, the values do not exceed 10%.

L39, 241, 247, 252, 258, 276 – Third instar nymph

A: We adopted the suggestions provided by R1. The nymphal stage has been specified.

OBS: After checking the data, we noticed that the columns in table 1 were mixed up and we corrected them. 

Reference number 1 has been updated.

---

## [Decision Letter · Decision Letter 2]

12 Mar 2023

From the dual cyclone harvest performance of single conidium powder to the effect of Metarhizium anisopliae on the management of Thaumastocoris peregrinus (Hemiptera: Thaumastocoridae)

PONE-D-22-33367R2

Dear Dr. Mauricio M Domingues, 

We’re pleased to inform you that your manuscript has been judged scientifically suitable for publication and will be formally accepted for publication once it meets all outstanding technical requirements.

Kind regards,

Shawky M Aboelhadid, PhD

Academic Editor

PLOS ONE

Additional Editor Comments (optional):

Reviewers' comments:

Reviewer's Responses to Questions

**Comments to the Author**

1. If the authors have adequately addressed your comments raised in a previous round of review and you feel that this manuscript is now acceptable for publication, you may indicate that here to bypass the “Comments to the Author” section, enter your conflict of interest statement in the “Confidential to Editor” section, and submit your "Accept" recommendation.

Reviewer #1: All comments have been addressed

2. Is the manuscript technically sound, and do the data support the conclusions?

Reviewer #1: Yes

3. Has the statistical analysis been performed appropriately and rigorously? 

Reviewer #1: Yes

4. Have the authors made all data underlying the findings in their manuscript fully available?

Reviewer #1: Yes

5. Is the manuscript presented in an intelligible fashion and written in standard English?

Reviewer #1: Yes

6. Review Comments to the Author

Reviewer #1: Dear editor, the manuscript is ready to be published.

Thank you for the opportunity to evaluate the manuscript.

7. PLOS authors have the option to publish the peer review history of their article (what does this mean?). If published, this will include your full peer review and any attached files.

Reviewer #1: No

<quillbot-extension-portal></quillbot-extension-portal>

---

## [Editor Report · Acceptance letter]

17 Mar 2023

PONE-D-22-33367R2 

From the dual cyclone harvest performance of single conidium powder to the effect of *Metarhizium anisopliae* on the management of *Thaumastocoris peregrinus* (Hemiptera: Thaumastocoridae) 

Dear Dr. Domingues:

I'm pleased to inform you that your manuscript has been deemed suitable for publication in PLOS ONE. Congratulations! Your manuscript is now with our production department. 

Kind regards, 

on behalf of

Professor Shawky M Aboelhadid 

Academic Editor

PLOS ONE